# Enriching Knowledge Bases with *Interesting* Negative Statements

**Hiba Arnaout**                                    HARNAOUT@MPI-INF.MPG.DE
**Simon Razniewski**                                SRAZNIEW@MPI-INF.MPG.DE
**Gerhard Weikum**                                  WEIKUM@MPI-INF.MPG.DE
*Max Planck Institute for Informatics, Germany*

## Abstract

Knowledge bases (KBs), pragmatic collections of knowledge about notable entities, are an important asset in applications such as search, question answering and dialogue. Rooted in a long tradition in knowledge representation, all popular KBs only store positive information, but abstain from taking any stance towards statements not contained in them.

In this paper, we make the case for explicitly stating interesting statements which are *not* true. Negative statements would be important to overcome current limitations of question answering, yet due to their potential abundance, any effort towards compiling them needs a tight coupling with ranking. We introduce two approaches towards automatically compiling negative statements. (i) In *peer-based statistical inferences*, we compare entities with highly related entities in order to derive potential negative statements, which we then rank using supervised and unsupervised features. (ii) In *pattern-based query log extraction*, we use a pattern-based approach for harvesting search engine query logs. Experimental results show that both approaches hold promising and complementary potential. Along with this paper, we publish the first datasets on interesting negative information, containing over 1.4M statements for 130K popular Wikidata entities.

## 1. Introduction

**Motivation and problem** Structured knowledge is crucial in a range of applications like question answering, dialogue agents, and recommendation systems. The required knowledge is usually stored in KBs, and recent years have seen a rise of interest in KB construction, querying and maintenance, with notable projects being Wikidata [Vrandečić and Krötzsch, 2014], DBpedia [Auer et al., 2007], Yago [Suchanek et al., 2007], or the Google Knowledge Graph [Singhal, 2012]. These KBs store positive statements such as *"Canberra is the capital of Australia"*, and are a key asset for many knowledge-intensive AI applications.

A major limitation of most of these KBs is their inability to deal with negative information. At present, the major KBs contain virtually only positive statements, whereas statements such as that *"Tom Cruise did not win an Oscar"* could only be inferred with the major assumption that the KB is complete - the so-called *closed-world assumption* (CWA). Yet as KBs are only pragmatic collections of positive statements, the CWA is not realistic to assume, and there remains uncertainty whether statements not contained in a KBs are false, or truth is merely unknown to the KB.

Not being able to formally distinguish whether a statement is false or unknown poses challenges in a variety of applications. In medicine, for instance, it is important to distinguish between knowing about the absence of a biochemical reaction between substances,

and not knowing about its existence at all. In corporate integrity, it is important to know whether a person was never employed by a certain competitor, while in anti-corruption investigations, absence of family relations needs to be ascertained. In data science and machine learning, on-the-spot counterexamples are important to ensure the correctness of learned extraction patterns and associations.

**State of the art and its limitations** Current web-scale KBs contain almost only positive statements, and this is engraved in the *open-world assumption* (OWA) employed on the semantic web, which states that asserted statements are true, while the remainder is unknown (a notable exception being Wikidata, which we discuss in details in Appendix E). Some formal entailment regimes like OWL [McGuinness et al., 2004] go beyond the OWA assumption, and allow to infer negation, yet are intended for use at query time, not for static materialization, and also lack ranking facilities. Similarly, data constraints [Marx and Krötzsch, 2017] and association rules [Ortona et al., 2018] can in principle yield negative statements, but face the same challenges.

This has consequences for usage of KBs: for instance, today's *question answering* (QA) systems are well geared for positive questions, and questions where exactly one answer should be returned (e.g., quiz questions or reading comprehension tasks) [Fader et al., 2014, Yang et al., 2015]. In contrast, for answering negative questions like *"Actors without Oscars"*, QA systems lack a data basis. Similarly, they struggle with positive questions that have no answer, like *"Children of Angela Merkel"*, too often still returning a best-effort answer even if it is incorrect. Materialized negative information would allow a better treatment of both cases.

**Approach and contribution** In this paper, we make the case that important negative knowledge should be explicitly materialized. We motivate this selective materialization with the challenge of overseeing a near-infinite space of possibly true statements that are not asserted in KBs, and with the importance of explicit negation in search and question answering. We then develop two complementary approaches towards generating negative statements: statistical ranking methods for statements derived based on related entities, and pattern-based text extraction, applied to high-quality search engine query logs. We also present the first datasets on interesting negative information, and highlight the usefulness of negative knowledge in extrinsic use cases.

Our salient contributions are:

1. We make the first comprehensive case for materializing interesting negative statements in KBs;
2. We present two judiciously designed methods for collecting negative statements: peer-based statistical inference and pattern-based text extraction;
3. We produce two datasets containing over 1.4M interesting negative statements for 130K popular Wikidata subjects.
4. We show the usefulness of negative knowledge in a QA use case.

## 2. Problem and Design Space

**Formalization** For the remainder we assume that a KB is a set of statements, each being a triple $(s; p; o)$ of subject $s$, property $p$ and object $o$.

Let $K^i$ be an (imaginary) ideal KB that perfectly represents reality, i.e., contains exactly those statements that hold in reality. Under the OWA, (practically) available KBs $K^a$ contain correct statements, but may be incomplete, so the condition $K^a \subseteq K^i$ holds, but not the converse [Razniewski and Nutt, 2011]. We distinguish two forms of negative statements:

**Definition 1 (Negative statements)**
  1. *A grounded negative statement $\neg(s; p; o)$ is satisfied if $(s; p; o)$ is not in $K^i$.*
  2. *A universally negative statement $\neg\exists(s; p; \_)$ is satisfied if there exists no o such that $(s; p; o) \in K^i$.*

Both statements represent standard logical constructs, and could also be expressed in the OWL ontology language. Grounded negative statements could be expressed via negative property statements (e.g., `NegativeObjectPropertyStatement(:hasWife :Bill :Mary)`), while universally negative statements could be expressed via `owl:complementOf` and `ObjectSomeValuesFrom` [Erxleben et al., 2014]. For these classes of negative statements, checking that there is no conflict with a positive statement is trivial. Yet compiling negative statements faces two other challenges. First, being not in conflict with positive statements is a necessary but not a sufficient condition for correctness of negation, due to the OWA. In particular, $K^i$ is only a virtual construct, so methods to derive correct negative statements have to rely on the limited positive information contained in $K^a$, or utilize external evidence, e.g., from text. Second, the set of correct negative statements is near-infinite, especially for grounded negative statements. Thus, unlike for positive statements, negative statement construction/extraction needs a tight coupling with ranking methods.

**Problem 1** *Given an entity e in a KB, compile a ranked list of interesting grounded negative and universally negative statements.*

**Design space** A first thought is that *deletions from time-variant KBs* are a natural source. For instance, in Wikidata, for subjects of type person within the last year, more than 500K triples have been deleted. Yet on careful inspection we found that most of these concern ontology restructuring, granularity refinements, or blatant typos, thus do not give rise to interesting negation.

A second conceivable approach is to leverage the CWA, or its relaxed variant PCA (Partial Completeness Assumption, aka. LCWA for Local CWA) [Galárraga et al., 2015], to generate negative statements. Using just the active domain of Wikidata for grounding, the CWA would give rise to about $6.4 \times 10^{18}$ negative statements[1]. Assuming that Wikidata covers 10% of all true statements per entity, more than 99.999% of the negative statements would be correct, but hardly interesting. For the PCA, the total would be about $3.2 \times 10^{16}$ negative statements (assuming an average of 5 populated properties per entity), and almost all of these would be correct. But these approaches would miss the true issue: merely enumerating huge sets of negative statements is not insightful even with (trivially) high precision. The key challenge rather is to identify interesting statements that users find noteworthy.

Instead, we propose methods that follow two major paradigms for KB construction and completion: *Statistical inference* and *text extraction*.

---

1. 80 Million subjects times 1000 properties times 80 Million objects.

Statistical inference methods, ranging from association rule mining such as AMIE and RuDiK [Galárraga et al., 2013, Ortona et al., 2018] to embedding models such as TransE and HolE [Bordes et al., 2013, Nickel et al., 2016] can predict positive statements and provide ranked lists of role fillers for KB relations. In Section 4, we develop a statistical inference method for negative statements, which generates candidate sets from related entities, and uses a set of popularity and probability heuristics in order to rank these statements.

Textual information extraction (IE) is a standard paradigm for KB construction. Common challenges in textual IE comprise noise and sparsity in observations, and canonicalization of entities and predicates. Our goal is to achieve maximal flexibility w.r.t. open predicates, and to overcome sparsity in negative statements in texts. Section 5 presents a method that combines pattern-based and open information extraction, and applies it to a particularly rich data source, search engine query logs.

## 3. Related Work

The problem of compiling informative negative statements about entities is new, so there are no directly comparable methods. Nevertheless, there is prior work on rule mining over KBs [Galárraga et al., 2015] that is conceivably useful in our context.

Most notably, [Galárraga et al., 2017] employed rule mining to predict the completeness of properties for given entities. This corresponds to learning whether the PCA holds in a local part of the KB, inferring that all absent values for a subject-predicate pair are false. For our task, this could be a building block, but it does not address the inference of *interesting* negative statements.

[Ortona et al., 2018] devised a rule mining system that can learn rules with negative atoms in rule heads (e.g., people born in Germany cannot be US president). This could be utilized towards predicting negative statements. Unfortunately, the mining also discovers many convoluted and exotic rules (e.g., people whose body weight is less than their birth year cannot win the Nobel prize), often with a large number of atoms in the rule body, and such rules are among the top-ranked ones. Even good rules, such as "people with birth year after 2000 do not win the Nobel prize", are not that useful for our task. Such rules predict way too many – correct, but uninformative – negative statements, essentially enumerating a huge set of people who are not Nobel laureates.

[Ortona et al., 2018] also proposed a precision-oriented variant of PCA that assumes negation only if subject and object are connected by at least one other relation. Unfortunately, this condition is rarely met in interesting cases. For instance, none of the negative statements in Table 2 have alternative connections between subject and object in Wikidata.

Another related line of work is learning which attributes are mandatory in a KB, for only non-mandatory absent predicates are candidates for universal absence. [Lajus and Suchanek, 2018] exploits density differences along type hierarchies to this end. This could be an initial filter towards discovering negative statements, but does not address our key problem of inferring when a missing statement is truly negative and interesting.

## 4. Peer-based inference

The first method combines information from similar entities ("peers") with supervised calibration of ranking heuristics. The intuition behind this method is that similar entities can give cues towards what expectations regarding relevant statements for an entity are. For instance, several entities similar to the physicist *Stephen Hawking* have won the Nobel prize in Physics. We may thus conclude that him not winning this prize could be an especially interesting statement. Yet related entities also share other traits, e.g., many famous physicists are US citizens, while Hawking is British. We thus need to devise ranking methods that take into account various cues such as frequency, importance, unexpectedness, etc.

**Peer-based candidate retrieval** To scale the method to web-scale KBs, in the first stage, we compute a candidate set of negative statements using the CWA, to be ranked in the second stage. Given a subject $e$, we proceed in three steps:
1. *Obtain peers:* We collect entities that set expectations for statements that $e$ could have, the so-called *peer groups* of $e$. Peer groups can be based (i) on structured facets of the subject [Balaraman et al., 2018], such as *occupation, nationality*, or *field of work* for humans, or classes/types for other entities, (ii) graph-based measures such as distance or connectivity [Ponza et al., 2017], or (iii) entity embeddings such as TransE [Bordes et al., 2013], possibly in combination with clustering, thus reflecting latent similarity.
2. *Count statements:* We count the relative frequency of all predicate-object-pairs (i.e., $(\_, p, o)$) and predicates (i.e., $(\_, p, \_)$ within the peer groups, and retain the maxima, if candidates occur in several groups. In this way, statements are retained if they occur frequently in at least one of the possibly orthogonal peer groups.
3. *Subtract positives:* We remove those predicate-object-pairs and predicates that exist for $e$.

The method's algorithm and an example are shown with more details in Appendix A.

**Ranking negative statements** Given potentially large candidate sets, in a second step, ranking methods are needed. Our rationale in the design of the following four ranking metrics is to combine frequency signals with popularity and probabilistic likelihoods in a *learning to rank model*.
1. *Peer frequency (PEER):* The statement discovery procedure already provides a relative frequency, e.g., 0.9 of a given actor's peers are married, but only 0.1 are political activists. The former is an immediate candidate for ranking.
2. *Object popularity (POP):* When the discovered statement is of the form $\neg(s; p; o)$, its relevance might be reflected by the popularity[2] of the *Object*. For example, $\neg(Brad Pitt; award; Oscar for Best Actor)$ would get a higher score than $\neg(Brad Pitt; award; London Film Critics' Circle Award)$, because of the high popularity of the Academy Awards over the latter.
3. *Frequency of the Property (FRQ):* When the discovered statement has an empty *Object* $\neg\exists(s; p; \_)$, the frequency of the *Property* will reflect the authority of the statement. To compute the frequency of a *Property*, we refer to its frequency in the KB. For example,

---

2. Wikipedia page views

$\neg\exists$(Joel Slater; citizen; _) will get a higher score (3.2M citizenships in Wikidata) than $\neg\exists$(Joel Slater; twitter; _) (160K twitter usernames).

4. *Pivoting likelihood (PIVO):* In addition to these frequency/view-based metrics, we propose to consider textual background information about $e$ in order to better decide whether a negative statement is relevant. To this end, we build a set of statement pivoting classifier [Razniewski et al., 2017], i.e., classifiers that decide whether an entity has a certain statement/property, each trained on the Wikipedia embeddings [Yamada et al., 2018] of 100 entities that have a certain statement/property, and 100 that do not[3]. To score a new statement/property candidate, we then use the pivoting score of the respective classifier, i.e., the likelihood of the classifier to assign the entity to the group of entities having that statement/property.

The final score of a candidate statement is then computed as follows.

**Definition 2 (Ensemble ranking)**

$$Score = \begin{cases} \lambda_1 PEER + \lambda_2 POP(o) + \lambda_3 PIVO & \text{if } \neg(s;\ p;\ o) \\ \lambda_1 PEER + \lambda_4 FRQ(p) + \lambda_3 PIVO & \text{if } \neg\exists(s;\ p;\ \_) \end{cases}$$

Hereby $\lambda_1, \lambda_2, \lambda_3$, and $\lambda_4$ are hyperparameters to be tuned on data withheld from training.

## 5. Pattern-based query log extraction

The second paradigm which we explore in this paper is text extraction. Text extraction comes with a space of choices for method and sources, in the method space most importantly distinguishing between supervised methods tailored to specific predicates, and unsupervised open information extraction. The former typically can reach higher precision, while the latter comes at greater flexibility towards unseen predicates.

For proof of concept, we thus opt here for an unsupervised method. To obtain negative statements, we use a few handcrafted *meta-patterns*, which we instantiate in the second step with entity mentions to retrieve textual occurrences.

Besides the extraction method, a crucial choice in textual IE is the text corpus. Beyond general topical coverage, typical design decision are whether to opt for larger, typically noisier text collections, or whether to focus efforts on smaller quality corpora with less redundancy. As proof of concept, we opt here for a small source of particularly high quality: search engine query logs, to which limited access can be obtained via autocompletion APIs [Romero et al., 2019].

**Meta-patterns** Inspired by work on identifying negated findings and diseases in medical discharge summaries [Chapman et al., 2001], we manually crafted 9 meta-patterns to retrieve negative information in query logs. All our meta-patterns start with the question word *"Why"*, because questions of this kind implicate that the questioner knows or believes the statement to be true, but wonders about its cause. We combine this question word with four kinds of negation, *n't, not, no and never*, which according to Blanco [Blanco and Moldovan, 2011] cover 97% of the explicit negation markers. Together with two tenses and two verb forms (have and do), these gave rise to a total of 9 frequent meta-patterns, listed

---

3. On withheld data, linear regression classifiers achieve 74% avg. accuracy on this task.

in Appendix B.

**Query log extraction**   Search engine query logs are normally a well-guarded secret of search engine providers. As proposed in [Romero et al., 2019], a way to probe their contents is to exhaustively query autocompletion APIs with strings with iteratively growing alphabetic prefixes, e.g., *"Why hasn't Stephen Hawking"*, *"Why hasn't Stephen Hawking a"*, *"Why hasn't Stephen Hawking b"*, and so on. The returned autocomplete suggestions then provide a glimpse into frequent queries to the platform. The returned queries are not yet representing statements, but questions. To turn them into the form of statements, we utilize ClausIE [Del Corro and Gemulla, 2013], obtaining for instance from the query *"Why didn't Stephen Hawking win the Nobel prize?"* the statement *(Stephen Hawking, did not win, the Nobel prize)*. The whole process is illustrated in Appendix B.

## 6. Experimental Evaluation

### 6.1 Peer-based Inference

We instantiate the peer-based inference method with 30 peers, popularity based on Wikipedia page views, and peer groups based on entity occupations. The choice of this simple peering function is inspired by Recoin [Balaraman et al., 2018]. In order to further ensure relevant peering, we also only considered entities as candidates for peers, if their Wikipedia viewcount was at least a quarter of that of the subject entity. We randomly sample 100 popular Wikidata humans. For each of them, we collect 20 negative statement candidates: 10 with the highest peer score, 10 being chosen at random from the rest of retrieved candidates. We then used crowdsourcing to annotate each of these 2000 statements on whether it was interesting enough to be added to a biographic summary text (Yes/Maybe/No). Each task was given to 3 annotators. Interpreting the answers as numeric scores (1/0.5/0), we found a standard deviation of 0.29, and full agreement of the 3 annotators on 25% of the questions. Our final labels are the numeric averages among the three annotations.

We tune the ranking model on data withheld from training (details in Appendix C.1), finding best parameter values -0.03 (*PEER*), 0.09 (*FRQ*), -0.04 (*POP*), 0.13 (*PIVO*), and a constant value of 0.3.

To compute the ranking quality of our method against a number of baselines, we use the Discounted Cumulative Gain (DCG) [Järvelin and Kekäläinen, 2002], which is a measure that takes into consideration the rank of relevant statements and can incorporate different relevance levels.

We use three *baselines*: As a naive baseline, we randomly order the 20 statements per entity. This baseline gives a lower bound on what any ranking model should exceed. We also use two competitive embedding-based baselines, TransE [Bordes et al., 2013] and HolE [Nickel et al., 2016]. We plug their prediction score for each candidate grounded negative statement.[4]

Table 1 shows the average $nDCG$ over the 100 entities for top-k negative statements for k equals 3, 5, 10, and 20. As one can see, our ensemble outperforms the best baseline by 6 to

---

4. Note that both models are not able to score statements about universal absence, a trait shared with the object popularity heuristic in our ensemble.

Table 1: Ranking metrics evaluation results for peer-based inference.

| Ranking Model | Coverage(%) | $nDCG_3$ | $nDCG_5$ | $nDCG_{10}$ | $nDCG_{20}$ |
|---|---|---|---|---|---|
| Random | 100 | 0.37 | 0.41 | 0.50 | 0.73 |
| TransE [Bordes et al., 2013] | 31 | 0.43 | 0.47 | 0.55 | 0.76 |
| HolE [Nickel et al., 2016] | 12 | 0.44 | 0.48 | 0.57 | 0.76 |
| Property Frequency | 11 | **0.61** | **0.61** | **0.66** | **0.82** |
| Object Popularity | 89 | 0.39 | 0.43 | 0.52 | 0.74 |
| Pivoting Score | 78 | 0.41 | 0.45 | 0.54 | 0.75 |
| Peer Frequency | 100 | **0.54** | **0.57** | **0.63** | **0.80** |
| Ensemble | 100 | **0.60** | **0.61** | **0.67** | **0.82** |

Table 2: Top-3 results for *Albert Einstein* using 3 ranking metrics.

| Random rank | Property frequency | Ensemble |
|---|---|---|
| $\neg\exists$(instagram; _) | $\neg\exists$(doctoral student; _) | $\neg$(occup.; astrophysicist) |
| $\neg$(child; Tarek Sharif) | $\neg\exists$(candidacy in election; _) | $\neg$(party; Communist Party USA) |
| $\neg$(award; BAFTA) | $\neg\exists$(noble title; _) | $\neg\exists$(doctoral student; _) |

16% in NDCG. The coverage column reflects the percentage of statements that this model was able to score. For example, for the *Popularity of Object* metric, a universally negative statement will not be scored. The same applies to TransE and HolE, for which we used pretrained models from [Ho et al., 2018]. Ranking with the *Ensemble* and ranking using the *Frequency of Property* outperformed all other ranking metrics and the three baselines, with an improvement over the random baseline of 20% for k=3 and k=5. Examples of ranked top-3 negative statements for *Albert Einstein* are shown in Table 2. That Einstein notably refused to work on the Manhattan project, and was suspected of communist sympathies is noteworthy. Also, despite his status as famous researcher, he truly never formally supervised any PhD student.

## 6.2 Correctness Evaluation

**Peer-based inference** We used crowdsourcing to assess the accuracy of results from the peer-based method. We collected 1K negative statements belonging to the three types used in the entity summarization experiments in Section 6.4. Every statement was annotated 3 times as either correct, incorrect, or ambiguous. 62% of the statements were found to be correct, 31% were incorrect, and 6% were ambiguous. Interpreting the scores numerical (0/0.5/1), annotations showed a standard deviation of 0.23. The annotated dataset is made available in Section 7.

**PCA vs. CWA** For a sample of 200 statements about humans (10 each for 20 entities), half generated only relying on the CWA, half additionally filtered to satisfy the PCA (subject has at least one other object for that property [Galárraga et al., 2015]), we manually checked correctness. We observed 84% accuracy for PCA-based statements, and 57% for CWA-based statements. So the PCA yields significantly more correct negative statements, though losing the ability to predict universal absence.

## 6.3 Pattern-based Query Log Extraction

Due to its coverage limitations, we focus the text extraction evaluation on the interestingness of extracted statements, not on ranking. We randomly sampled 100 popular humans from Wikidata for which our method could produce at least 3 negative statements *expressible* in Wikidata. For each of these entities, we collect their top-3 negative statements using five methods: our pattern-based query log extraction method (*QLE*), our method but with only Wikidata expressible properties (*QLE-canonicalized*), our peer-based inference method with the *Ensemble* ranking metric, TransE [Bordes et al., 2013], and HolE [Nickel et al., 2016]. We submit the retrieved statements to crowdworkers and we ask them whether they found each statement interesting enough to add it to a biographic summary text (Yes/Maybe/No). Results show the average relevance over the 100 entities for top-3 negative statements. The average relevance is 65, 61, 69, 77, and 77%, for TransE [Bordes et al., 2013], HolE [Nickel et al., 2016], *peer-based-ensemble*, *QLE*, and *QLE-canonicalized*. Our pattern-based query log extraction method, in both versions, outperforms the three baselines by 8, 12, and 16 percentage points. Further details on this experiment are in Appendix C.2.

## 6.4 Extrinsic Evaluation

We next highlight the relevance of negative statements for *entity summarization*. A second extrinsic use case, question answering, is discussed in Appendix D.

In this experiment we analyze whether mixed positive-negative statement set can compete with standard positive-only statement sets in the task of entity summarization. In particular, we want to show that the addition of negative statements will *increase the descriptive power* of structured summaries.

We collect 100 Wikidata entities from 3 diverse types: 40 humans, 30 organizations (including publishers, financial institutions, academic institutions, cultural centers, businesses, and more), and 30 literary works (including creative work like poems, songs, novels, religious texts, theses, book reviews, and more). On top of the negative statements that we infer, we collect relevant positive statements about those entities.[5] We then compute for each entity $e$ a sample of 10 positive-only statements, and a mixed set of 7 positive and 3 *correct*[6] negative statements, produced by the peer-based method. We rely on peering using Wikipedia embeddings [Yamada et al., 2018]. Annotators were then asked to decide which set contains more new or unexpected information about $e$. More particularly, for every entity, we ask workers to assess the sets (flipping the position of our set to avoid biases), leading to a total number of 100 tasks for 100 entities. We collect 3 opinions per task. Overall results show that mixed sets with negative information were preferred for 72% of the entities, sets with only positive statements were preferred for 17% of the entities, and the option "both or neither" was chosen for 11% of the entities. Table 4 shows results per each considered type. The standard deviation is 0.24, and the percentage of queries with full agreement is 22%. Table 3 shows three diverse examples. The first one is *Daily Mirror*. One particular interesting negative statement in this case is that the newspaper is not owned by the "News UK" publisher which owns a number of of British newspapers like The

---

5. We define a number of common/interesting properties to each of type, e.g., for humans, "position held" is a relevant property for positive statements.

6. We manually check the correctness of these negative statements

Table 3: Results for the entities *Daily Mirror*, *Peter the Great*, and *Twist and Shout*.

| Daily Mirror | |
|---|---|
| **Only-pos** | **Pos-and-neg** |
| (owned by; Reach plc) | ¬*(newspaper format; broadsheet)* |
| (newspaper format; tabloid) | (newspaper format; tabloid) |
| (country; United Kingdom) | ¬*(country; United States of America)* |
| (language of work or name; English) | (language of work or name; English) |
| (instance of; newspaper) | ¬*(owned by; News UK)* |
| ... | ... |

| Peter the Great | |
|---|---|
| **Only-pos** | **Pos-and-neg** |
| (military rank; general officer) | (military rank; general officer) |
| (owner of; Kadriorg Palace) | (owner of; Kadriorg Palace) |
| (award; Order of the Elephant) | ¬*(place of death; Moscow)* |
| (award; Order of St. Andrew) | (award; Order of St. Andrew) |
| (father; Alexis of Russia) | ¬*(award; Knight of the Order of St. Alexander Nevsky)* |
| ... | ... |

| Twist And Shout | |
|---|---|
| **Only-pos** | **Pos-and-neg** |
| (composer; Phil Medley) | ¬*(composer; Paul McCartney)* |
| (performer; The Beatles) | (performer; The Beatles) |
| (producer; George Martin) | ¬*(composer; John Lennon)* |
| (instance of; musical composition) | (instance of; musical composition) |
| (lyrics by; Phil Medley) | ¬*(lyrics by; Paul McCartney)* |
| ... | ... |

Table 4: Only-pos vs. pos-and-neg statements.

| **Preferred Choice** | Human (%) | Organization (%) | Literary work (%) |
|---|---|---|---|
| pos-and-neg | 71 | 77 | 66 |
| only-pos | 22 | 10 | 17 |
| both or neither | 7 | 13 | 17 |

Times, The Sunday Times, and The Sun. The second entity is *Peter the Great* who died in Saint Petersburg and not Moscow, and who did not receive the Order of St Alexander Nevsky which was first established by his wife, a few months after his death. And the third entity is *Twist and Shout*. Although it is a known song by The Beatles, they were not its composers, writers, or original performers.

## 7. Discussion

**Experiment results** Peer-based inference significantly outperformed the baseline methods, and property frequency was the single most important feature, indicating that uni-

Table 5: Negative statements for *Theresa May.*

| Query log | Peer-based inference |
|---|---|
| (not invited; Prince Harry's wedding) | $\neg\exists$(child; _) |
| (does not want; another referendum) | $\neg$(occup.; Economist) |
| (does not have; a deputy prime minister) | $\neg\exists$(sibling; _) |

Table 6: Negative statements for hotels in the United States.

| Hotel | Price | Room features | Hotel features |
|---|---|---|---|
| **Hotel Americas** | expensive | $\neg$ minibar; $\neg$ sofa; $\neg$ kitchenette | $\neg$ free-Wifi; $\neg$ pets; $\neg$ free-parking |
| **Scottish Inn & Suites** | budget | $\neg$ seating-area; $\neg$ iron; $\neg$ safety-box | $\neg$ fitness-center; $\neg$ bar; $\neg$ business-facilities |

versally negative statements are generally much more interesting than grounded negative statements. The two presented methods are instances of very different paradigms, consequently the question arises how they compare. As major differences we note that text extracted statements were found to be 8% more interesting, while peer-based inference is easier to apply to long-tail entities. We exemplify results from the two methods side-by-side in Table 5. Both methods present initial proof of concepts, with limitations in recall. As established in positive inference and text, we can easily trade off higher recall for somewhat lower precision, depending on the downstream use case. We also expect that recall can be boosted by tapping additional data sources, like structured data from tables and lists, as well as unstructured text.

**Relevance to other domains** Negative statements are highly important also in specific domains. In online shopping, characteristics not possessed by a product, such as the IPhone 7 not having a headphone jack, are a frequent topic highly relevant for decision making. The same applies to the hospitality domain: the absence of features such as free WiFi or gym rooms are important criteria for hotel bookers, although portals like Booking.com currently only show (sometimes overwhelming) positive feature sets. To illustrate this, Table 6 shows interesting negative features of standard rooms of major hotels in the U.S., as per their listing on Booking.com, using the peer-based method on 50 comparable hotels. Although some of these may simply represent data issues, information such as that the Scottish Inn & Suites does not offer a safety box may provide important cues for decision making.

**Negative statement datasets for Wikidata** [7] We publish the first datasets that contain dedicated negative statements about entities in Wikidata: (i) Peer-based statistical inference data: 1.4M negative statements about the most popular 130K people, organizations, and literary works, and (ii) Pattern-based query log extraction data: 6.2K negative statements about the most popular 2.4K people. We also release the mturk-annotated 1k negative statements of Sec. 6.2.

**Demo**[7] A web-based browsing interface will be made accessible.

---

7. https://www.mpi-inf.mpg.de/departments/databases-and-information-systems/research/knowledge-base-recall/interesting-negations-in-kbs/

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

# Appendices

## A. Details on Peer-based Inference

Algorithm 1 shows the full procedure of the peer-based inference method. In line 1, peers are selected based on some blackbox function *peer_groups*. Subsequently, for each peer group, we collect all statements and properties that these peers have, and rank them by their relative frequency. Across peer groups, we retain the maximum relative frequencies, if a property or statement occurs across several. Before returning the top results as output, we subtract those already possessed by entity $e$.

---

**Algorithm 1:** Peer-based candidate retrieval algorithm.

**Input** : knowledge base KB, entity $e$, peer group function $peer\_groups$, size of a group of peers $s$, number of results $k$

**Output:** $k$-most frequent negative statement candidates for $e$

2  $\boldsymbol{P}[] = peer\_groups(e, s)$ ;                                // collecting peer groups

3  $N[] = $ ;                                // final list of scored negative statements

4  **for** $P_i \in \boldsymbol{P}$ **do**

5    $L\_candidates = []$ ;                  // predicate and predicate-object pairs of group $P_i$

6    $uL\_candidates = []$ ;                                // unique values of $L\_candidates$

7    **for** $pe \in P_i$ **do**

8      $L\_candidates += collect(pe, p, \_)$ ;                  // pe: peer, p: predicate

9      $L\_candidates += collect(pe, p, o)$ ;                                // o: object

10   **end**

11   $uL\_candidates = unique(L\_candidates)$

12   **for** $st \in uL\_candidates$ **do**

13     $sc = \frac{count(st, L\_candidates)}{s}$ ;                  // scoring statements, st: statement

14     **if** $getscore(st, N) < sc$ **then**

15       $setscore(st, sc, N)$

16     **end**

17   **end**

18 **end**

19 $N -= inKB(e, N)$ ;                                // remove statements $e$ already has

20 **return** $max(N, k)$

---

Moreover, an example is shown in Table 7 for $e=Brad\ Pitt$. In this example, we instantiate the peer group choice to be based on structured information, in particular, shared occupations with the subject, as in Recoin [Balaraman et al., 2018]. In Wikidata, Pitt has 8 occupations *(actor, film director, model, ...)*, thus we would obtain 8 peer groups of entities sharing one of these with Pitt. For readability, let us consider statements derived from only one of these peer groups, *actor*. Let us assume 3 entities in that peer group, *Russel Crowe, Tom Hanks, and Denzel Washington*. The list of negative candidates, $L\_candidates$, are all the predicate and predicate-object pairs shown in the columns of the 3 actors. And in this particular example, $N$ is just $uL\_candidates$ with scores for only the "actor" group, namely (award; Oscar for Best Actor):1.0, (citizen; New Zealand):0.33, (child; _):1.0, (occupation; screenwriter):1.0, (convicted; _):0.33, and (citizen; U.S.A.):0.67. Positive candidates of *Brad Pitt* are then dropped from $N$, namely (citizen; U.S.A.):0.67 and (child; _):1.0. The top-k of the rest of candidates in $N$ are then returned. For k=3 for example, the top-k negative statements are ¬(award; Oscar for Best Actor), ¬(occupation; screenwriter), and ¬(citizen; New Zealand).

Note that without proper thresholding, the candidate set grows very quickly, for instance, if using only 30 peers, the candidate set for Brad Pitt on Wikidata is already about 1500 statements.

Table 7: Discovering candidate statements for *Brad Pitt* from one peer group with 3 peers.

| Russel Crowe | Tom Hanks | Denzel Washington | Brad Pitt | Candidate statements |
|---|---|---|---|---|
| (award; Oscar for Best Actor) | (award; Oscar for Best Actor) | (award; Oscar for Best Actor) | (citizen; U.S.A.) | ¬(award; Oscar for Best Actor), 1.0 |
| (citizen; New Zealand) | (citizen; U.S.A.) | (citizen; U.S.A.) | (child; _) | ¬(occup.; screenwriter), 1.0 |
| (child; _) | (child; _) | (child; _) | | ¬(citizen; New Zealand), 0.33 |
| (occup.; screenwriter) | (occup.; screenwriter) | (occup.; screenwriter) | | ¬∃(convicted; _), 0.33 |
| (convicted; _) | | | | |

Figure 1: Retrieving negated statements about an entity e from text.

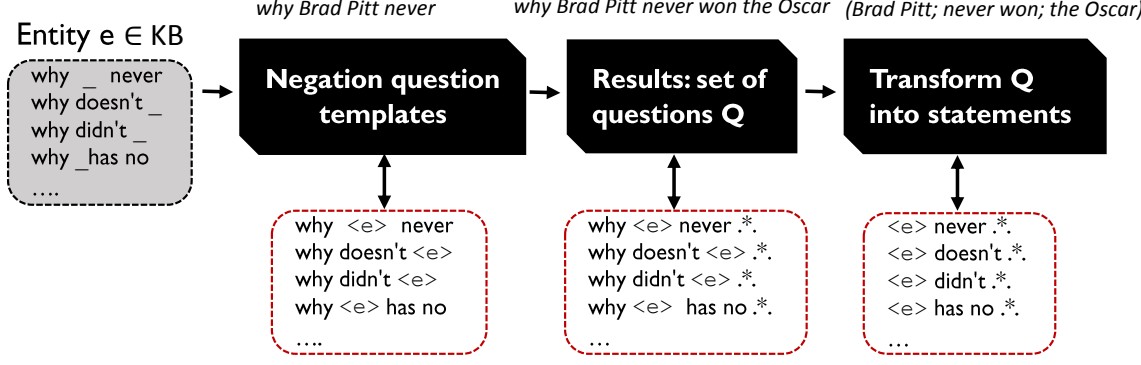

## B. Details on Pattern-based Query Log Extraction

Figure 1 illustrate the process of retrieving interesting negative statements about an entity *e* from a given search engine's query log. The meta patterns used in this method are shown in Table 8.

## C. Experiment Details

### C.1 Ensemble Ranking: Hyperparameters Tuning

To learn optimal hyperparameters for the ensemble ranking function (Definition 2), we trained a linear regression model using 5-fold crossvalidation on the 2000 labels for interestingness. Four example rows are shown in Table 9. Note that the ranking metrics were normalized using a ranked transformation to obtain a uniform distribution for every feature.

Table 8: Meta patterns.

| Meta-pattern | Freq. (%) | Meta-pattern | Freq. (%) |
|---|---|---|---|
| Why isn't <e> | 35 | Why hadn't <e> | 3 |
| Why didn't <e> | 28 | Why <e> has no | 2 |
| Why doesn't <e> | 21 | Why wasn't <e> | 1 |
| Why <e> never | 6 | Why <e> had no | 1 |
| Why hasn't <e> | 3 | | |

Table 9: Data samples for illustrating hyperparameter tuning.

| Statement | PEER | FRQ(p) | POP(o) | PIVO | Lab. |
|---|---|---|---|---|---|
| ¬(Bruce Springsteen; award; Grammy Lifetime Achievement Award) | 0.8 | 0.8 | 0.55 | 0.25 | 0.83 |
| ¬(Gordon Ramsay; lifestyle; mysticism) | 0.3 | 0.8 | 0.8 | 0.65 | 0.33 |
| ¬∃(Albert Einstein; doctoral student; _) | 0.85 | 0.9 | 0.15 | 0.4 | 0.66 |
| ¬∃(Celine Dion; educated at; _) | 0.95 | 0.95 | 0.25 | 0.95 | 0.5 |

The average obtained optimal hyperparameter values were -0.03 for *Peer Frequency*, 0.09 for *Frequency of Property*, -0.04 for *Popularity of Object*, and 0.13 for *Pivoting likelihood*, and a constant value of 0.3., with a 71% out-of-sample precision.

## C.2 Pattern-based Query Log Extraction: Evaluation

We randomly sampled 100 popular humans from Wikidata, for which our method could produce at least 3 negative statements expressible in Wikidata. For example, the statement *(Brad Pitt, never won, Oscar for Best Actor)* can be transformed into the Wikidata statement ¬*(Brad Pitt; award received; Oscar for Best Actor)*, with the property `P166`. Popular humans are defined as humans with Wikipedia page views higher than the average page views for entities of type human. For each of these entities, we collected their top-3 negative statements using five methods: our pattern-based query log extraction method (*QLE*), our method but with only Wikidata expressible properties (*QLE-canonicalized*), our peer-based inference method with the *Ensemble* ranking metric, TransE [Bordes et al., 2013], and HolE [Nickel et al., 2016]. For *QLE-canonicalized*, we collect the 30 most frequent properties in the dataset we publish in Section 7, that can be expressed in Wikidata. The properties that can be expressed with Wikidata properties are mapped manually. We replace them in the collected set of statements by replacing the property with the Wikidata property and adding the ¬ symbol to the beginning of the statement. For the former two methods, the source of the data is the query log, for the third it is Wikidata, and for the latter two it is a subset of Wikidata (300K statements) containing prominent entities of different types [Ho et al., 2018], which we enriched with all facts about the sampled entities. We submit the retrieved statements to crowdworkers to answer 4500 tasks (5 methods, 100 entities, 3 statements/entity, 3 judgments/statement). We ask the annotators whether they found each statement interesting enough to add it to a biographic summary text (Yes/Maybe/No). Interpreting the answers as numeric scores (1/0.5/0), we found a standard deviation of 0.2, and full agreement of the three annotators on 29% of the questions. Our final labels are the numeric averages among the three annotations.

Moreover, to validate the correctness of query log extraction, we sampled another 100 random human entities from the top 3K most popular humans in Wikidata. We retrieved all the negative statements for them, and annotated a sample of 100 statements along two dimensions: (i) Correctness (correct/ambiguous/incorrect), (ii) Wikidata-expressivity. The latter captures whether the statement could be expressed as a single triple by use of an existing Wikidata property (e.g., *"Paul Mccartney is not vegan"* can be expressed in Wikidata via `P1576`), whether the predicate currently has no corresponding Wikidata property, but its existence is conceivable (e.g., *"Albert Einstein did not drive."*), or whether the statement is too subjective or complex to be sensible for a KB (e.g., *"Madonna does*

*not like Lady Gaga"*). Results showed that 60% of the statements are correct, 22% are ambiguous, and only 18% are incorrect. We also found that 36% are *KB-expressible*, 26% are *expressible with new property*, and 38% are *inexpressible*.

## D. Extrinsic Evaluation in Question Answering

In this experiment we compare the results to negative questions over a diverse set of sources. We manually compiled five questions that involve negation, such as *"Actors without Oscars"*[8]. We compare them over a four highly diverse sources: Google Web Search (increasingly returning structured answers from the Google knowledge graph [Singhal, 2012]), WDAqua [Diefenbach et al., 2017] (an academic state-of-the-art KBQA system), the Wikidata SPARQL endpoint (direct access to structured data), and our peer-based inference method. For Google Web Search and WDAqua, we submit the queries in their textual form, and consider answers from Google if they come as structured knowledge panels. For Wikidata and peer-based inference, we transform the queries into SPARQL queries[9], which we either fully execute over the Wikidata endpoint, or execute the positive part over the Wikidata endpoint, while evaluating the negative part over a dataset produced by our peer-based inference method. For each method, we then self-evaluate the number of results, the correctness and relevance of the (top-5) results.

All methods are able to return highly correct statements, yet Google Web Search and WDAqua provide no answers to answer 3 and 2 of the queries at all. Wikidata SPARQL returns by far the highest number of results, 250K on average, yet does not perform ranking, thus returns results that are hardly relevant (e.g., a local Latvian actor to the Oscar question). The peer-based inference outperforms it by far in terms of relevance (72% vs. 44% for Wikidata SPARQL), and we point out that although Wikidata SPARQL results appear highly correct, this has no formal foundation, due to the absence of a stance of OWA KBs towards negative knowledge.

## E. Negation in Wikidata

Wikidata's [Vrandečić and Krötzsch, 2014] ability to express some kinds of negation is a notable exception among major KBs. We discuss these next.

**Universally Absent Negative Statements.** Wikidata can capture statements about *universal absence* via the "no-value" symbol. This allows KB editors to add a statement where the object is empty. For example, what we express as $\neg\exists$(Angela Merkel; child; _), the current version of Wikidata allows to be expressed as (Angela Merkel; child; no-value)[10]. As of last year, there exist 122K of such "no-value" statements, yet only used in narrow domains. For instance, 53% of these statements come for just two properties "country" (used almost exclusively for geographic features in Antarctica), and "follows" (indicating that an artwork is not a sequel).

---

8. Textual queries: "actors with no Oscars", "actors with no spouses", "film actors who are not film directors", "football players with no Ballon d'Or", "politicians who are not lawyers".

9. SPARQL queries: w.wiki/A6r, w.wiki/9yk, w.wiki/9yn, w.wiki/9yp, w.wiki/9yq

10. https://www.wikidata.org/wiki/Q567

**Count Predicates.** Another way of expressing negation is via counts matching with instances, for instance, adding 5 children of Trump, and the statement "number_of_children=5". Yet Wikidata lacks a principled approach for dealing with these, especially, only few of these count predicates exist, and there is no formal link with the corresponding regular enumerating predicate [Ghosh et al., 2020].

**Negated Predicates.** Wikidata contains a few relations that carry a negative meaning, for instance it *does not have part* (243 statements), or *different from* (492K statements). Yet these present very specific pieces of knowledge, e.g., (arm; does not have part; hand), (Hover Church; does not have part; bell tower), and (brain death; different from; death) which do not generalize to other Wikidata properties.

**Deprecation of Statements.** One possible way of expressing *ground negative statements* is by using the deprecated rank feature. Even though this allows editors to flag a statement as negative, and allows them to provide a reason for the deprecation, it it mostly used for statements discovered as incorrect during the editing process, instead of explicitly and actively adding notable negative information, like that "Stephen Hawking did not win the Nobel Prize in Physics."

**Ongoing Discussions.** An interesting discussion took place on the *Wikidata's Project Chat* webpage[11] about the need for an "opposite" to a property. More particularly, the opposite of the property *complies with (P5009)* to state when an entity does not comply with the criterion associated with an entity. There is a way of stating that the film "Beauty and the Beast" *complies with* the "Bechdel test". However, stating that the film "Hackers" fails to comply with the "Bechdel test" cannot be done by simply negating the property *complies with*, but through a workaround that introduced the negative entity (object) *"fails the Bechdel Test (Q45172088)"*, and then stating that "Hackers" *has quality "fails the Bechdel Test (Q45172088)"*. This, however, is not a practical nor generalizable way to deal with every possible negation that the KB presents.

---

11. https://www.wikidata.org/wiki/Wikidata:Property_proposal/fails_compliance_with