# OpenReview forum: "Enriching Knowledge Bases with Interesting Negative Statements"
_AKBC.ws/2020/Conference — AKBC 2020_

### Official Review · AnonReviewer3 · 2020-03-15
**Very important problem, but I am not sure the paper addresses it**

**Rating:** 5
**Confidence:** 3

**Review:**

This paper studies how interesting negative statements can be identified for knowledge bases (KBs). The main contributions are several ideas of how to generate negative statements, and several heuristics to rank them.

This paper sets foot into a much-needed domain of research. Negative statements are a very important issue for today's KBs, and the paper does not just formalize the problem and propose means to generate and rank such statements, but also provides user studies. The video of the demo is particularly impressive!

I have three main issues with this submission: First, the methods seem to be geared exclusively to famous entities, and more specifically to famous humans. The peer-ranking works great for "Which actors won the academy award", but might work much less well on "Which villages do not have a mayor". The Google auto-completion, likewise, works great for "Which football players won the Ballon d'Or", but it is less clear how it works for "Which classical musical pieces are not written in B flat major" (assuming that there is such a Wikidata relation). Thus, the paper should more accurately be called "enriching KBs with interesting negative statements about famous humans".

The second issue is more fundamental: If I understand correctly, the peer-ranking method makes the closed world assumption. It computes the attributes that peers of the target entity have, and that the target entity itself does not have. These are proposed as negative statements. However, these statements are not necessarily false -- they may just be missing from the KB. The evaluation of the method ignores that problem: It asks users to rate the negative statements based on interestingness -- but does not give the user the option to say "This statement is actually not false, it is true". In this way, the proposed method ignores the main problem: that of distinguishing missing information from wrong information. That is surprising, because the paper explicitly mentions that problem on page 9, complaining that the Wikidata SPARQL approach has no formal foundation due to the open world assumption. It is just the same with the first proposed approach. This is what the paper itself states on Page 10: “Textual evidence is generally a stronger signal that the negative statement is truly negative” --  implying that the first proposed method does not always produce correct negative statements. However, the main part of the paper does not acknowledge this or evaluate whether the produced negative information is actually negative.

Only in the appendix (which is optional material), we find that the peer-based method has a "71% out-of-sample precision", which, if it were the required value, should be discussed in the main paper. The same goes for the value of 60% for the query-log-based method.

The third issue is the evaluation: The proposed methods should be explicitly evaluated wrt. their correctness (i.e., whether they correctly identify wrong statements) in the main part of the paper. Then, they should be compared to the baseline, which is the partial completeness assumption. This is currently not done.

The next question is how to rank the negative statements. The baseline here should be the variant of the partial completeness assumption that is used in RUDIK [Ortona et al, 2018]: It limits the partial completeness assumption to those pairs of entities that are connected in the KB. It says: "If r(x,y) and r'(x,z) and not r(x,z), then r(x,z) is an interesting negative statement". The proposed method should be compared to this competitor.

Thus, while the paper opens a very important domain of research, I have the impression that it oversells its contribution: by ignoring the question of missing vs. wrong statements, by not comparing to competitors, and by focusing its methods exclusively on famous humans.

Related work:
- The "universally negated statements" are the opposite of the "obligatory attributes" investigated in [2], which thus appears relevant.
- The work of [1] solves a similar problem to the submission: by predicting that a subject s has no more objects for a relation r than those of the KB, it predicts that all other s-r-o triples must be false.
- It appears that the peer-ranked method is a cousin of the partial completeness assumption of AMIE, and of the popularity heuristics used in [1]. It says: "If the KB creators took the care to annotate all your peers with this attribute, and if you had that attribute, they would for sure have annotated you as well. Since they did not, you do not have this attribute." This is a valid and very interesting method to generate negative statements, but it would have to be stated explicitly and evaluated for correctness.

Minor:
- It would be great to know the weights of the scores in Definition 2 also in the main paper.
- Talking of "text extraction" in Section 6 is a bit misleading, because it sounds as if the data was extracted from full natural language text, whereas it is actually extracted from query logs.
- It would be good to clarify how the Booking.com-examples were generated (manually or with the proposed method).

[1] Predicting Completeness in Knowledge Bases, WSDM 2017
[2] Are All People Married? Determining Obligatory Attributes in Knowledge Bases, WWW 2018

---

> ### Author Response · Authors · 2020-04-09
> **Thank you for the comments. We have revised parts of the paper.**
>
> Thanks for the comments. We have revised parts of the paper, with changes marked in blue.
>
> Entity scope:
> We have expanded the extrinsic evaluation in Section 6.4, for the task of entity summarization, from 5 entities of type human to 100 diverse entities (40 humans, 30 organizations, 30 artworks). Examples are shown in Table 3.
> As for prominent vs. long-tail entities, we added a discussion in Section 7 (Experimental Evaluation).
>
> Role of CWA and PCA:
> We do indeed rely on CWA among groups of notable peers. We clarified this in the paper.
> PCA is a reasonable reference point, too. Therefore, we added an experimental comparison of the precision for CWA and PCA in Section 6.2. PCA produces results with significantly higher accuracy than CWA. Thanks for making this suggestion in the review.
>
> We emphasize that the main contribution of our method lies in identifying *interesting* negative statements; we are not just adding to the vast literature on truth prediction. We revised the Design Space section (Section 2) by better clarifying that predicting correct negative statements alone misses out on the key point.
>
> Comparison with PCA:
> We added a post-hoc evaluation of filtering by PCA (Section 6.2).
>
> Positioning against RuDiK:
> We added a new section on Related Work (Section 3) where we discuss RuDiK and related approaches.
>
> More different types of entities:
> The web interface can now handle 10 types (including organizations and artworks). The demo video shows examples: https://bit.ly/39Cn2ES.
>
> Minor comments:
> We added a discussion of references [1] and [2] suggested by the reviewer in a new section on related work (Section 3).
> We move the discussion of weights for the ensemble ranking into the main body in Section 6.1.
> We clarified in Section 7 that the Booking.com examples were generated with the proposed peering method, using 50 manually chosen comparable hotels as reference.

---

### Official Review · AnonReviewer1 · 2020-03-27
**Official Blind Review #1**

**Rating:** 3
**Confidence:** 3

**Review:**

The paper addresses the problem of negative statements in knowledgebase. They formalize the types of negative statements: (a) grounded statement: [s,p,o] does not exist in KB, (b) not exist [s, p, o] (there's no object that satisfy s,p]. To find negative statements, they proposes two methods (a) peer-based candidate retrieval (i.e., heuristic of finding relation that is frequently populated in nearby entities but missing in the target entity) and (b) Using search logs with meta patterns (i.e., search query logs for pattern such as "Why XXX not", and find retrieved queries such as "Why XXX never won the Oscar).

I agree with the motivation behind this work — studying negative statement is a problem worth pursuing, especially to build a high precision QA system that does not hallucinate. Having said that, I have multiple concerns with the current version of the paper.

(1) Evaluation is not rigorous. Both extrinsic evaluation (entity summarization, question answering) is very small scale. Both evaluation only on five examples. I would rather preferred the paper to focus on one evaluation, but do the study much more carefully and in larger scale, reporting statistical significant and so forth.

(2) The notion of "Interesting" is very subjective. The paper does not even try to define what counts as an "interesting" negative statement. Is it highly likely fact that is not true? Does it mean that it is surprising and unknown?

(3) In section 5, What is nDCG? I don't think it is defined, and I don't know what Table 5 is talking about.

(4) The paper releases the negative statement datasets. I think this could be very valuable to the community if it is released with manual annotations, even for a small subset (2-3K examples). As is, this is model prediction that we don't have a good sense of accuracy, so not very useful.

Minor point:

- In section 5.2, it talks about randomly sampling 100 popular humans. What's the definition of "popular"? In the sentence afterwards, it talks about "expressible" in Wikidata. Does it mean it involves predicate can be mapped to Wikidata by string matching? by manual matching?
- In Section 4, what's the difference between "popularity" and "frequency"?
- It would be interesting to see the actual values for hyperparameters for Definition 2. What do you mean by "withheld training data"?

---

> ### Author Response · Authors · 2020-04-09
> **Thank you for your helpful comments. We have revised parts of the paper.**
>
> Thanks for the comments. We have revised parts of the paper, with changes marked in blue.
>
> Most importantly, we have expanded the extrinsic evaluation in Section 6.4, for the task of entity summarization, from 5 entities of type human to 100 diverse entities (40 humans, 30 organizations, 30 artworks). Examples are shown in Table 3.
>
> Notion of interestingness:
> We agree that considering specific tasks would make the notion of interestingness more tangible. As KBs like Wikidata etc. are not built for one specific task, though, our annotation guidelines pursued a middle stance and grounded interestingness in an abstract task: “Imagine you are writing a biography about X. Is this statement interesting enough to be added to the biography?”. For the hotels (Section 7) it is easier to phrase a task like booking decisions.
>
> Definition of NDCG:
> We added the definition of the NDCG ranking metric on page 7. We clarified that Table 5 (now Table 7) exemplifies the peer-based inference with a toy example.
>
> Correctness of negative statements:
> We asked the crowd to manually annotate a subset of 1000 statements for correctness, discussed in Section 6.2, with data accessible at https://bit.ly/2V0eaUo .
>
> Minor comments:
> We ground “popularity” in the number of views of the Wikipedia page of a given entity. An entity is considered popular if its number of views are greater than or equal to the average number of views of entities of the same type. Frequency of a property/predicate is defined as the number of statements in the KB including this property.
> We moved the discussion of weights for the ensemble ranking into the main body in Section 6.1. “Withheld training data” was corrected to “data withheld from training”, that is, we properly split the data into train/validation/test.
> In Section 6.3: we now clarify in Appendex C.2 that “popular humans” means humans with Wikipedia page views higher than the average page views for entities of type human. We also clarify in Appendix C.2 that the mapping of Wikidata expressible properties is done manually for the 30 most frequent properties in our result set.

---

### Official Review · AnonReviewer2 · 2020-03-28
**Well-motivated work in a direction worth exploring**

**Rating:** 7
**Confidence:** 4

**Review:**

This paper studies constructing interesting negative statements about entities to enrich knowledge bases. The authors propose two main approaches and evaluate using both crowdsourcing and extrinsic evaluations on entity summarization and question answering.

Pros:
- I really like the idea of adding negative statements and the paper provides good motivations for why these are necessary for different domains and downstream tasks. I think this work lays a good starting point for a line of follow-up studies.
- The authors explore two approaches from the two main regimes to generate interesting negative statements about entities utilizing some heuristics and do experiments to show their respective weaknesses and strengths.
- Dataset collected is of large scale and can be potentially be used for learning tasks on interesting negative statements.

Cons:
- The extrinsic evaluations seem a bit synthetic and small-scale. It would be interesting to see how actually enriching a KB using these negative statements could help, for example, solving a large open-domain QA dataset.
- More complicated baselines could be included such as recent transformer-based language models on the open-domain QA evaluation.

In summary, I think this paper is well-written, well-motivated, and lays a good starting point for an important omitted direction in KB-related research.

---

> ### Author Response · Authors · 2020-04-09
> **Thank you for your comments, we have revised parts of the paper.**
>
> Thanks for the comments. We have revised parts of the paper, with changes marked in blue. Most importantly, we have expanded the extrinsic evaluation in Section 6.4, for the task of entity summarization, from 5 entities of type human to 100 diverse entities (40 humans, 30 organizations, 30 artworks). Examples are shown in Table 3.

---

### Official Review · AnonReviewer3 · 2020-04-17
**Great work**

**Rating:** 9
**Confidence:** 3

**Review:**

I am posting here my review from before the revision of the paper by the authors. All my concerns have been addressed in that revision.



This paper studies how interesting negative statements can be identified for knowledge bases (KBs). The main contributions are several ideas of how to generate negative statements, and several heuristics to rank them.

This paper sets foot into a much-needed domain of research. Negative statements are a very important issue for today's KBs, and the paper does not just formalize the problem and propose means to generate and rank such statements, but also provides user studies. The video of the demo is particularly impressive!

I have three main issues with this submission: First, the methods seem to be geared exclusively to famous entities, and more specifically to famous humans. The peer-ranking works great for "Which actors won the academy award", but might work much less well on "Which villages do not have a mayor". The Google auto-completion, likewise, works great for "Which football players won the Ballon d'Or", but it is less clear how it works for "Which classical musical pieces are not written in B flat major" (assuming that there is such a Wikidata relation). Thus, the paper should more accurately be called "enriching KBs with interesting negative statements about famous humans".

The second issue is more fundamental: If I understand correctly, the peer-ranking method makes the closed world assumption. It computes the attributes that peers of the target entity have, and that the target entity itself does not have. These are proposed as negative statements. However, these statements are not necessarily false -- they may just be missing from the KB. The evaluation of the method ignores that problem: It asks users to rate the negative statements based on interestingness -- but does not give the user the option to say "This statement is actually not false, it is true". In this way, the proposed method ignores the main problem: that of distinguishing missing information from wrong information. That is surprising, because the paper explicitly mentions that problem on page 9, complaining that the Wikidata SPARQL approach has no formal foundation due to the open world assumption. It is just the same with the first proposed approach. This is what the paper itself states on Page 10: “Textual evidence is generally a stronger signal that the negative statement is truly negative” --  implying that the first proposed method does not always produce correct negative statements. However, the main part of the paper does not acknowledge this or evaluate whether the produced negative information is actually negative.

Only in the appendix (which is optional material), we find that the peer-based method has a "71% out-of-sample precision", which, if it were the required value, should be discussed in the main paper. The same goes for the value of 60% for the query-log-based method.

The third issue is the evaluation: The proposed methods should be explicitly evaluated wrt. their correctness (i.e., whether they correctly identify wrong statements) in the main part of the paper. Then, they should be compared to the baseline, which is the partial completeness assumption. This is currently not done.

The next question is how to rank the negative statements. The baseline here should be the variant of the partial completeness assumption that is used in RUDIK [Ortona et al, 2018]: It limits the partial completeness assumption to those pairs of entities that are connected in the KB. It says: "If r(x,y) and r'(x,z) and not r(x,z), then r(x,z) is an interesting negative statement". The proposed method should be compared to this competitor.

Thus, while the paper opens a very important domain of research, I have the impression that it oversells its contribution: by ignoring the question of missing vs. wrong statements, by not comparing to competitors, and by focusing its methods exclusively on famous humans.

Related work:
- The "universally negated statements" are the opposite of the "obligatory attributes" investigated in [2], which thus appears relevant.
- The work of [1] solves a similar problem to the submission: by predicting that a subject s has no more objects for a relation r than those of the KB, it predicts that all other s-r-o triples must be false.
- It appears that the peer-ranked method is a cousin of the partial completeness assumption of AMIE, and of the popularity heuristics used in [1]. It says: "If the KB creators took the care to annotate all your peers with this attribute, and if you had that attribute, they would for sure have annotated you as well. Since they did not, you do not have this attribute." This is a valid and very interesting method to generate negative statements, but it would have to be stated explicitly and evaluated for correctness.

Minor:
- It would be great to know the weights of the scores in Definition 2 also in the main paper.
- Talking of "text extraction" in Section 6 is a bit misleading, because it sounds as if the data was extracted from full natural language text, whereas it is actually extracted from query logs.
- It would be good to clarify how the Booking.com-examples were generated (manually or with the proposed method).

[1] Predicting Completeness in Knowledge Bases, WSDM 2017
[2] Are All People Married? Determining Obligatory Attributes in Knowledge Bases, WWW 2018

---

### Public Comment · ~Christian_Kleineidam1 · 2020-05-25
**Wikidata can store negative statements**

I'm an admin over at Wikidata.

This paper claims in it's introduction that Wikidata is currently unable to store negative claims. In reality Wikidata does store negative statements by giving negative statements the rank deprecated. Since 2015 it's possible to qualify the nature of the deprecation with reason for deprecation.

Wikidata for example stores the claim that Bill Henry (https://www.wikidata.org/wiki/Q4909403) was falsly believed to have died in September 2007 and we know it's false because he was seen afterwards.

The paper claims that Wikidata struggle to give meaningful answers to questions about children of Angela Merkel. At the time of the writing of this paper Wikidata was able to tell a reader that Angela Merkel has 0 children. Wikidata can additionally store "novalue" for the query that asks to list Angela Merkels children.

The data that's produced by the interference seems to be low quality as it conflates negative statements with unknown statements. If any researcher wants to pursue further work at creating a dataset of interesting negative statements, that dataset would likely be welcome on Wikidata and I invite any researcher to approach us at Wikidata on https://www.wikidata.org/wiki/Wikidata:Project_chat .

---

> ### Author Response · Authors · 2020-05-28
> **Thanks for pointing this out!**
>
> Dear Christian:
>
> Thanks for pointing this out! We now address the Wikidata case in more details as a separate section (Appendix E).
>
> The fact that Wikidata can represent universal absence via no-values is great and we are sorry to have missed that. We will add this point to the paper.
>
> You are also right that for the special case of Merkel, the fact that she has no children could be alternatively retrieved from the special property “number_of_children=0”. Yet that doesn’t generalize, e.g., there are no “number_of_citizenships” or “number_of_teams” properties.
>
> Concerning the deprecation though, we believe, it is intended for flagging older incorrect statements, as part of the KB curation process. The incorrect statements are still kept as versions or variants, in line with the Wikidata rationale to capture alternative claims as well.
> In contrast, our goal is to identify *interesting* negated statements that should be explicitly added to a KB as salient information (e.g., Hawking did not win a Nobel prize).
>
> The observation that the output of our method contains both truly negative statements and incorrect statements (i.e., claimed negative albeit true in reality) is right. However, no computational method can realistically guarantee that it yields only truly negative statements. Our ranking methods aim to generate strong candidates that do indeed fall in the realm of negative statements and are noteworthy. This could be an asset that helps KB curators. We would welcome feedback from the Wikidata community, and would gladly engage ourselves in follow-up work.
>
> Best wishes,
> Hiba, Simon, Gerhard

---

### Decision · Program_Chairs · 2020-05-01

**Decision:**

Accept

**Comment:**

This paper explores a new direction in knowledge base construction: how to identify *interesting* negative statements for KBs.  Towards this general goal, two approaches have been developed: peer-based statistical inference and pattern-based text extraction. Two datasets of negative knowledge bases are provided, along with an extrinsic QA evaluation.

There has been quite a bit of discrepancy among the reviews. All the reviewers appreciated that this paper addresses a very important (and previously underestimated) problem but there are lots of discussion around the evaluation: (1) whether the current evaluation is too small-scale/non-rigorous, (2) whether the closed-world assumption is reasonable or not, (3) the correctness of evaluation of extracted KBs.

The authors have made substantial revisions during the rebuttal phase and we believe most of these issues have been addressed. Therefore, we recommend accepting this paper.